# Sentinel Lymph Node Biopsy in Cutaneous Melanoma, a Clinical Point of View

**DOI:** 10.3390/medicina58111589

**Published:** 2022-11-03

**Authors:** Daciana Elena Brănişteanu, Mihai Cozmin, Elena Porumb-Andrese, Daniel Brănişteanu, Mihaela Paula Toader, Diana Iosep, Diana Sinigur, Cătălina Ioana Brănişteanu, George Brănişteanu, Vlad Porumb, Alin Constantin Pînzariu, Sorin Liviu Băilă, Alin Codruț Nicolescu

**Affiliations:** 1Department Dermatology, ‘Grigore T. Popa’ University of Medicine and Pharmacy, 700115 Iasi, Romania; 2Railway Clinical Hospital, 700506 Iasi, Romania; 3Clinical Department, “Apollonia” University of Iasi, 700511 Iasi, Romania; 4Department of Ophthalmology, ‘Grigore T. Popa’ University of Medicine and Pharmacy, 700115 Iasi, Romania; 5‘Grigore T. Popa’ University of Medicine and Pharmacy, 700115 Iasi, Romania; 6Department of Surgery, ‘Grigore T. Popa’ University of Medicine and Pharmacy, 700115 Iasi, Romania; 7Department of Morpho-Functional Sciences II, Discipline of Physiology, ‘Grigore T. Popa’ University of Medicine and Pharmacy, 700115 Iasi, Romania; 8Vascular Surgery Group, Ponderas Academic Hospital, 014142 Bucharest, Romania; 9Roma Medical Center for Diagnosis and Treatment, 011773 Bucharest, Romania

**Keywords:** cutaneous melanoma, sentinel lymph node biopsy, clinical guidelines

## Abstract

Sentinel lymph node biopsy (SLNB) is a surgical procedure that has been used in patients with cutaneous melanoma for nearly 30 years. It is used for both staging and regional disease control with minimum morbidity, as proven by numerous worldwide prospective studies. It has been incorporated in the recommendations of national and professional guidelines. In this article, we provide a summary of the general information on SLNB in the clinical guidelines for the management of cutaneous malignant melanoma (American Association of Dermatology, European Society of Medical Oncology, National Comprehensive Cancer Network, and Cancer Council Australia) and review the most relevant literature to provide an update on the existing recommendations for SLNB.

## 1. Introduction

Cutaneous melanoma (CM) is a malignant neoplasm that develops from melanocytes and is one of the most lethal forms of skin malignancy [1]. Melanoma is a major public health problem because of its rapid rise in incidence over the last 50 years, high mortality, complexity, and treatment costs, particularly in advanced stages [2,3,4]. Sentinel lymph node biopsy (SLNB), first reported in 1992 by Morton and Cochran, has since become the standard of care in the management of early-stage melanoma patients and allows the assessment of lymph node status. Sentinel lymph node (SLN) is the first lymph node (LN) on a direct drainage channel from the primary tumor to the regional nodal basin that is closest to the site of the primary melanoma. It is because lymphatic drainage follows a similar anatomical path, with tumor cells draining straight into one or more lymph nodes [5,6]. Preoperative and intraoperative lymphatic mapping are performed, followed by selective lymphadenectomy of the first LN detected along the lymphatic drainage pathway from the primary tumor to the regional nodal basin. With a high degree of accuracy, this method identifies the LN most likely to harbor any cellular metastases from the primary tumor, detecting clinically occult diseases [5,7,8]. The prognostic importance of SLN status is now well-established and widely acknowledged [9]. SLNB detects clinically occult nodal metastases in 20% of patients at the time of diagnosis, making it the most important independent predictor of survival. As a result, SLNB improves the outcomes of patients with an occult disease by preventing clinical regional node involvement and identifies patients with pathologic node-negative diseases who will not benefit from CLND (completion LN dissection) [6,10,11]. For the standard assessment and management of patients with cutaneous malignant melanoma, professional societies such as the American Academy of Dermatology, the European Society for Medical Oncology, the National Cancer Institute, and the Cancer Council of Australia have developed guidelines.

### SLNB Technique

SNLB is a minimally invasive surgical method that dramatically changed the surgical staging of CM. The SLN status, or the absence or presence of subclinical nodal metastases in the first lymph nodes to receive tumor cells from the original location, is assessed with this approach. The most important prognostic factor in early-stage melanoma patients is the disease status of regional LNs [6,12,13,14]. This enables the clinician to identify individuals who are at a higher risk of recurrence and may benefit from further treatments, especially when adjuvant medications are used [9,15,16]. The current practice of SNLB involves preoperative lymphatic mapping (lymphoscintigraphy) around the primary CM or biopsy scar, the injection of 99mTc labeled radiopharmaceutical with the possibility of using the SPECT/CT hybrid imaging, and intraoperative SLN localization using a handheld gamma probe with or without the use of blue dye. For the SLNB procedure to be accurate, it is of critical importance that all “true” SLNs are identified and removed for examination [6].

Both routine histology and immunohistochemistry are used to assess SLNs for the presence of tumor involvement [17,18]. SLNB allows a precise histologic examination of the entire regional nodal basin, which can be used to diagnose or exclude a clinically occult disease. SLNB has dramatically improved the prognosis of patients with metastatic melanoma to regional LNs compared to clinical examination or imaging tests. The presence of a positive SLN is the best predictive factor for recurrence and survival, and it increases with an increasing initial tumor thickness, ulceration, and other adverse clinicopathologic prognostic factors [6,11,19]. SLN positivity divides intermediate and high-risk primary melanomas into subgroups with better or worse overall prognosis, making it easier to identify patients who would benefit from adjuvant therapies and, in some cases, additional surgery such as CLND [11,19,20,21,22]. The SLNB technique is based on the concept that lymphatic drainage from the site of cutaneous melanoma is an “orderly” process, initially affecting only one or a few nodes. Therefore, an SLN is characterized as a lymphatic node that receives lymphatic drainage directly from the original tumor and serves as a first-line filter for tumor cells that have detached from the primary lesion. The remaining LNs in the basin are unlikely to harbor disease if the SLN is histologically negative for metastases. Thus, by analyzing the lymph node that is most likely to contain metastases, appropriate LN staging can be performed, perhaps avoiding the potential consequences of CLND. Furthermore, the SLNB procedure enhances the identification of nodal micrometastases by directing the pathologist to examine fewer LNs more thoroughly, allowing clinicians to identify patients with occult nodal metastases that would otherwise take months or years to become clinically palpable. As a result, despite SLNB’s less invasive nature, staging data obtained from one SLNB can be more accurate than those obtained from a CLND [6,23,24,25,26].

## 2. Importance, Limitations, and Complications of SLNB

### 2.1. Importance

Melanoma is staged using the eighth edition of the *American Joint Committee on Cancer* (AJCC) system. At the time of diagnosis, around 84% of the CM cases are presented with localized disease (AJCC stage I–II), 9% are involved with regional LNs (AJCC stage III), and 4% are presented with distant metastases (AJCC stage IV). The presence or absence of nodal disease, even micrometastases, has been found to be the most important prognostic factor in patients with early-stage melanoma (AJCC stage I–II) [6].

As melanoma metastases to the regional LNs are the most common sites of initial metastasis (AJCC stage III) [17], their early detection allows for early control of the regional disease. However, if left untreated, nodal micrometastases may develop into macrometastases, which ultimately and theoretically may promote the development of distant metastatic disease (AJCC IV stage) [22,27]. SLNB is the standard criterion for LNs staging in patients with CM. Pathologic staging of regional LNs identifies patients with CM and occult metastasis, immediately upstaging them to AJCC stage III. In patients with CM, SLN status (positive or negative) is regarded as the most important prognostic factor for recurrence and the best predictor of survival [17]. Because physical examination and imaging modalities are insufficiently sensitive to detect the occult (microscopic) disease, a histopathological assessment of LNs is necessary for the early detection of melanoma metastasis. SLNB provides excellent information for melanoma staging, establishing prognosis, and considering further therapy [6].

Accurate LN staging with minimal potential risks is possible by analyzing the LN that is most likely to contain metastases. CLND was used to stage clinically node-negative melanoma patients until the introduction of SLNB in the early 1990s. As a result, many patients have been exposed to unwarranted risks of short- and long-term lymphedema, hematoma, seroma, wound infection, nerve dysfunction, discomfort, functional deficiency, and swelling because of CLND. Furthermore, CLND has not been found to improve survival in multiple prospective trials. SLNB has been validated over time and is now widely acknowledged as the standard approach for staging clinically localized cutaneous melanoma in the TNM system [28,29].

### 2.2. Limitations

First, the primary goal of SLNB is to surgically remove and analyze only true SLNs, but its accuracy depends on the correct visualization and identification of these true SLNs. Second, it is vital to avoid crucial errors that could result in the retention of LNs carrying metastatic cells, which could proceed to clinically detectable disease. SLNB false-negative rates have been reported to range between 5.6 percent and 21 percent, with local LN recurrence following a negative SLNB being the most common indicator. As a result, interdisciplinary collaboration between radiological, surgical, and pathological departments, as well as physician expertise, is required to avoid a false-negative SLN [9].

On the other hand, the success of SLN identification techniques is greatly due to strict uniformity, and technical features are crucial for correct nodal staging. The true biological SLN is the node with the greatest chance of harboring metastases, not the one nearest to the tumor, the most visible on preoperative imaging, or even the most radioactive in the surgery field [8,30,31]. Furthermore, even when the same body region is examined, lymphatic drainage from the skin differs significantly from one patient to another. Lymphoscintigraphy is required for reliable SLNB, especially in patients with head and neck melanomas, which are technically more difficult due to the complex and less predictable lymphatic circulation. For better anatomical localization of the SLN, international guidelines recommend using 3D imaging with SPECT/CT for preoperative planning and intraoperative decision making. This method has a greater detection rate and more precise SLN(s) localization and should be used in primary tumors in locations with difficult anatomy, as well as in melanoma patients with unexpected drainage on planar images, such as the trunk and head-neck region [32,33,34].

### 2.3. Complications

Because the complication rates are low (5–10%) and much lower than total LN dissection, SLNB is considered a less invasive surgical procedure [15,16]. A total of 88% of patients undergoing SLNB will have a negative result, with an average pooled rate of 11% experiencing a complication [35,36,37]. Lymphedema, seroma, hematoma, wound dehiscence, infection, lymphedema, or rarer consequences such as nerve injury, thrombophlebitis, deep vein thrombosis, hemorrhage, and anaphylaxis to blue dyes are all possible side effects and complications. During surgery, some of the lymph veins going to and from the SLN or adjacent group of nodes are cut, disrupting normal lymph flow, and causing swelling due to a build-up of lymphatic fluid in that and nearby areas. The lymph collection in the affected area may produce discomfort or pain over time, and the overlying skin may thicken or stiffen. Because the risk of lymphedema rises with the number of LNs removed, simply removing the SLN carries a lower risk. The injured body part may be difficult to move because of the discomfort and edema, and at the incision site, numbness or tingling, as well as infection or bruising, may appear [9,37].

## 3. Indications and Contraindications

### 3.1. Indications

Recommendations for SLN biopsy in CM are uniform across every major guideline developed by worldwide organizations and are consistent in interpreting its value and limitations [13,16,38,39]. In cases of localized CM without clinically detectable LNs, SLNB is the most accurate staging method. In the primary lesion, the main variables for risk of SLN metastasis are Breslow thickness, ulceration, and the number of mitoses. In general, if a patient’s risk of a positive SLN is <5% (MIS, T1a, nonulcerated lesions <0.8 mm in Breslow thickness), NCCN does not recommend SLNB; if a patient’s risk of a positive SLN is 5–10% (T1b melanoma, 0.8 to 1.0 mm in Breslow thickness or <0.8 mm in Breslow thickness with ulceration), NCCN recommends discussing and considering SLNB; if the probability of a positive SLN is >10% (T2a–T4b melanoma, (>1.0 mm in Breslow thickness), NCCN recommends that SLNB should be discussed and offered [15]. For patients with stage Ib or stage II melanoma, SLNB may occasionally be used as an inclusion criterion in treatment studies; however, this approach has generated debate. Even though SNLB has not helped to increase overall survival, it is a crucial tool for prognosis and staging and helps identify patients who might gain from adjuvant therapy. Evidence on prognosis after SLNB is limited for individuals with thick melanoma and the elderly. In practically all major cancer centers and melanoma programs across the world, SLNB (>1 mm recommended, 1 mm but stage Ib available) has taken the place of standard care for early-stage melanoma [11,40,41,42].

A new staging approach that redefined the T1 category of CM was published in the eighth edition of the American Joint Committee on Cancer (AJCC) cancer staging manual. With this new categorization, tumors were separated into T1a or T1b groups based on Breslow thickness and ulceration. Because thickness and ulceration alone may identify a T1b with a worse prognosis, a mitotic rate ≥ 1/mm^2^ was dropped from the staging system. Additionally, rather than reporting Breslow thickness to the nearest 0.01 mm, the eighth edition of the AJCC advised the reporting to the nearest 0.1 mm. As a result, tumors with a Breslow thickness of 0.8 mm or less and no ulceration were labeled as T1a, while tumors with a thickness of 0.8 mm or less and ulceration or with a thickness between 0.8 and 1.0 mm, were labeled as T1b [38].

In conclusion, SLNB should be considered in CM patients with a Breslow thickness of at least 1 mm, as well as for those with a thickness of 0.8 to 1.0 mm (with or without ulceration), or less than 0.8 mm, in the presence of risk factors that increase the likelihood of SLN positivity: young age (patients under the age of 40 typically have higher rates of SLN positivity than do patients over the age of 40), lymphovascular invasion, positive deep biopsy margin (if close to 0.8 mm), high mitotic rate (≥1 mitosis/mm^2^), or a combination of these factors [43,44,45]. There is debate on how to manage positive regional LNs in melanoma. Previous recommendations advised performing a CLND in all patients with a positive SLNB. However, individuals with a positive SLNB and monitored by nodal ultrasonography had comparable melanoma-specific survival and overall survival when compared to those with a CLND, according to the landmark melanoma SLNB trials MSLT-II and DeCOG-SLT. The Multicenter Selective Lymphadenectomy Trial (MSLT)-II compared CLND versus active ultrasound nodal observation in patients with a positive SLNB and found that, at a median follow-up of 43 months, CLND did not improve melanoma-specific survival in all patients with SLN metastasis but did immediately increase the rate of regional disease control and staging among patients with a positive SLN. CLND vs. active nodal basin ultrasound monitoring should be considered and provided in positive SLNB, according to the National Comprehensive Cancer Network’s (NCCN) most recent guidelines [16,46].

### 3.2. Contraindications

Other factors to consider when deciding whether to perform SLNB in melanoma patients include advanced age, poor functional status, known local or systemic disease spread, prior extensive surgery in the immediate area of the primary tumor or of the targeted lymph node basin, and comorbidities that could result in a short life expectancy or prevent general anesthesia or further treatment [5,16]. As people get older, their lymphatic systems become more variable, making SLNs harder to detect, and SLNB accuracy decline as well [47]. Although SLNB may be technically more challenging and have lower prognostic value in older patients, there is currently no agreement on an upper age cutoff that would advise against this technique [17].

Furthermore, patients who have primary melanoma and satellitosis or “in-transit” metastases should not be given SLNB because they are already in AJCC stage III. The information provided by SLNB will not change the prognosis or the course of treatment [14]. While blue dyes must be avoided, radioactive colloid tracers can be used safely with dose adjustments without affecting accuracy while doing SLNB during pregnancy. However, it should be avoided for a few days after SLNB to breastfeed [9,15].

## 4. Current Role of SLNB in the Management of Cutaneous Melanoma

Following these recommendations will not always guarantee a favorable outcome. The decision on whether a certain therapy is appropriate or not should be made by the doctor and the patient after considering all the facts that each patient has presented, as well as the known biological variation and behavior of the disease [17]. The choice to do SLNB should always be tailored to the patient and discussed with them while considering the risks and advantages of this intervention and the patient’s particular needs and preferences [9]. Three important randomized controlled trials: Multicenter Selective Lymphadenectomy Trial I (MSLT-I), German Dermatologic Cooperative Oncology Group-Selective Lymphadenectomy Trial (DeCOG-SLT), and Multicenter Selective Lymphadenectomy Trial II (MSLT-II) have transformed the era of surgery for regional LNs in melanoma. A total of 1347 stage I/II melanoma patients eligible for SLNB participated in the multicenter, randomized MSLT-I trial and were randomly assigned to one of two groups. MSLT-I compared SLN biopsy versus no SLN biopsy in patients with localized CM and assessed the impact of SLNB on the survival of CM patients. Patients in the first group underwent comprehensive melanoma excision combined with SLN biopsy. The patients underwent immediate CLND if the SLN was positive. Patients with negative SLNs were followed up. Patients in the second group received wide excision alone and were closely monitored; CLND was only advised if nodal metastases were clinically visible during the follow-up. Overall, the two groups did not differ significantly in melanoma-specific survival. The data showed that SLN histological status was the most significant prognostic factor for the survival of patients with localized CM (with clinically negative LNs) and that SLNB offers superior disease-free survival (DFS) and has switched from a therapeutic to a prognostic role [18,45].

The German Dermatologic Cooperative Oncology Group–Sentinel Lymph node Trial (DeCOG-SLT was the first randomized clinical trial to assess the benefit of complete lymphadenectomy (CL) in melanoma patients with positive SLN biopsy. Enrolled in this trial were 483 patients with cutaneous melanoma on the trunk and limbs with a median follow-up of 35 months. No difference in metastasis-free survival was found between the groups with “dissected” nodal chain and the groups with spared nodal chain and followed by trimonthly ultrasound (66% of cases had micrometastases < 1.0 mm in the SLN). The authors concluded that CL should not be performed if the SLN presents micrometastasis less than 1.0 mm [48,49].

Patients in the MSLT-II study had a positive SLN detected by histology or RT-PCR and were randomly assigned to receive CLND immediately or observation with frequent clinical evaluation (clinical examination with associated ultrasound assessment). Melanoma-specific survival was the main goal, while disease-free survival and non-SLN involvement were the secondary goals. The trial found no significant differences in melanoma-specific survival (MSS). However, the disease-free survival was slightly higher in the dissection arm. The authors concluded that CL, after SLN biopsy results were positive, could be waived, especially to spare patients from lymphedema, as it had no effect on melanoma-specific survival, particularly in patients with little nodal deposit in the SLN and who were willing to undergo stringent ultrasound follow-up. Along with MSLT-1, a few other larger studies also showed fewer surgical complications (24% vs. 41%) and that SLNB followed by immediate CLND resulted in better nodal disease control (reduced tumor burden) than delayed CLND. The number of involved nodes, surgical failure rates, the extent of extranodal tumor spread, surgical complication rates, and postoperative morbidity were all raised when CLND was delayed. CLND is no longer recommended because of its associated morbidity and lack of survival benefit in melanoma patients who receive treatment [46,50,51,52,53,54]. These studies and widespread practice in most institutions have led to the recommendation that SLN biopsy should be performed in melanomas, especially thin melanomas whenever there is a risk of a positive sentinel node greater than or equal to 5%. Based on these studies, it seems possible to identify a population of patients who might be qualified for the safe exclusion of SLN biopsy when the positive rate is less than 5% based on primary tumor depth (especially T1 and T2 melanomas), age, mitotic rate, lymphovascular invasion, and ulceration [20]. SLNB is being discussed in thin melanomas despite this group’s low reported rates of SLN positivity [11,15,16]. In thin melanomas, the risk of a positive SLN is reported to be 5%, compared to 15% for intermediate-thickness melanomas and 40% for thick melanomas [54,55,56,57,58,59]. In subcategories of thin melanoma in which the risk of metastases is >5%, SLNB is typically advised. The actual challenge is defining these high-risk subcategories because SLN-positive thin melanoma has been inconsistently correlated with high-risk parameters other than Breslow thickness. Selecting which of these patients would benefit from SLNB is, therefore, challenging [60,61,62,63,64,65,66,67,68]. According to most published research findings, in thick melanoma patients, SLNB should be suggested to stratify patients’ prognoses and determine which ones might benefit from new adjuvant therapies. As in thin melanomas, picking the right patients for SLNB is extremely wise. Recent research by Boada et al. suggests that thick nonulcerated lentigo malignant melanoma (LMM) and thick melanomas of other uncommon histological subtypes/other uncommon histological types of thick melanoma may not require SLNB. In fact, there is debate concerning the use of SLNB in individuals with pure desmoplastic melanomas [9,65,66,67]. The T1 category of CM was reclassified in the eighth edition of the American Joint Committee on Cancer (AJCC). National and international guidelines have been changed in accordance with the new AJCC staging method, but with slight variations. The new AJCC T1b patients should receive SLNB, whereas T1a patients should not, according to the amended ASCO/SSO guidelines [69]. A recent consensus in the United Kingdom recommended consideration of SLNB (to be considered) for all T1b patients, especially if lymphovascular invasion and a mitotic rate of less than 2/mm^2^ are present [70]. According to the most recent National Comprehensive Cancer Network (NCCN) guidelines, SLNB should also be considered in T1a melanoma patients with lymphovascular invasion, mitotic rate ≥ 2/mm^2^, or a combination of these two. Another reason to consider an SLNB is a case of partial biopsy of the original tumor with a positive deep margin near the 0.8 mm threshold. The American Academy of Dermatology’s (AAD) recommendations for treating melanoma are in line with NCCN recommendations: SLNB should be discussed and considered in T1a patients if other high-risk histological features are present, the patient is young (<40 years), or the primary tumor biopsy is insufficient or incomplete. In contrast, the current European Interdisciplinary Guideline on Melanoma does not recommend SLNB in patients with Breslow thickness below 0.8 mm and only does so if additional high-risk characteristics are present. Similar rules apply in Canada, the Italian Society of Dermatology and Venereology guidelines, and the French Cutaneous Oncology Group [71,72].

The updated guidelines of the Romanian Society of Dermatology and Venereology uses these recommendations (Figure 1).

In exceptional circumstances, such as very large melanomas, acromic/achromic/acral melanomas, and subungual melanomas, a skin biopsy may be performed by removing part of the lesion, known as an incisional, partial, or incomplete diagnostic biopsy, or it may be performed with the intention of removing the entire lesion (excisional or complete). In an emergency, the dermatologist, plastic surgeon, or oncology surgeon can do a skin biopsy (maximum of 10 days). After the excision, the histological and immunohistochemical results will be available in no more than two weeks, and the plastic surgeon or oncology surgeon will also do the SLNB at the same time. The oncologist develops the treatment plan in the following 10 days considering these findings. The patient is followed up by a dermatologist or oncologist at 1–3–6 months for 5 years.

Early-stage CM patients might receive care from a dermatologist (dermoscopy, follow-up at 1–6 months, and laboratory tests), oncologist (PET CT, appropriate medication, adjuvant therapy, visits scheduled every 3 or 6 months), and other specialists (as appropriate). The patients in advanced (metastatic) stages are monitored using PET CT, and laboratory tests, and receive advanced therapy.

Compared to thicker CM, the use of SLNB in T1 CM is more debatable, and research studies are still being conducted to identify the tumors that are more likely to be SLN-positive. Breslow thickness, which is the best predictor of SLN positivity, especially at or above the 0.75 mm (now 0.8 mm) threshold, is used by the NCCN recommendations to stratify consideration of SLNB in T1 CM. The Working Group recommends discussing SLNB with patients diagnosed with T1b CM, which is defined by the eighth edition of the AJCC staging system as less than 0.8 mm with ulceration or 0.8 to 1.0 mm with or without ulceration, even though overall SLN positivity rates in this subset of patients are still quite low (≤10%). SLN-positive rates in T1a CM (0.8 mm without ulceration) are typically less than 5%. Thus, the WG does not advise SLNB for individuals in the T1a subgroup unless other unfavorable histologic features are clearly present [17].

Even though histologic ulceration, lymphovascular invasion, and high mitotic rate (the threshold for which is yet unknown) are generally uncommon in T1 CM, they have all been linked to an increased chance of SLN positivity in several studies, but not all of them. Clinical decision-making is expected to be influenced by later investigations that evaluate the mitotic rate throughout its continuum for survival-based endpoints and define a pertinent cut-off for taking SLNB into account in T1 melanoma [17].

## 5. Psychological and Psychiatric Impact of Melanoma

In Romania, psychological support for cancer patients is not available in any oncology hospital, perhaps due to a large number of reported cases and the staff shortage in this field. Moreover, there are very few centers for diagnosing and treating these patients in Romania. Thus, healthcare workers are overworked; consequently, less attention is given to each case.

There is a need to integrate compensated psychiatry and psychotherapy services in treating cancer patients. When considering the needs and complaints of oncological patients, it is imperatively necessary for psychosocial support services to be available and compensated. Their inclusion among the compensated health services would improve the quality of life of these patients and prevent mental decompensation [73].

A cancer diagnosis is devastating for patients and often a traumatic event for them and their families. Most of the time, patients feel the need for psychological and even psychiatric counseling, which is only possible outside an oncology hospital, which entails physical effort and additional financial burden [73,74].

The first reaction to a cancer diagnosis is quite brutal. That is precisely why it would be advisable to have a multidisciplinary team for this information transfer. In the current conditions, the dermatologist or plastic surgeon wears surgical attire and not infrequently are not able to properly converse and give moral support to the patients and their families. Melanoma-induced depression and anxiety resolve shortly after melanoma patients undergo surgery (melanoma excision). However, one-in-ten melanoma patients develop new depression and anxiety symptoms, while in one in twenty, these symptoms persist. Chronic stress is responsible for the progression of melanoma, and melanoma patients should also receive mental health care to improve the outcomes of dermatological and oncological treatment. A frequent diagnosis in psychiatric practice is a mood disorder, depression type. It is characterized by depression symptoms and is caused by the patient’s oncological condition. The major psychological impact, surgical stress, and fear related to the traumatic aspects of chemotherapy (such as hair and significant weight loss, effects on all body systems with great physical and psychological impact) negatively influence patients’ moods. That is precisely why many of these patients, who also have sensitive personalities and for whom the road to convalescence was not easy, end up needing mental health help [75].

## 6. Conclusions

Years of experience have demonstrated that precise SLN mapping is essential to allow the surgical excision of only the true SLN(s) and, thus, to receive all therapeutic benefits of SLNB in terms of staging, prognosis, disease-free survival, and overall survival.

SLN biopsy should be performed with the appropriate technical expertise to correctly identify the sentinel node in the context of recognizing both the likelihood of positivity in a specific patient and the prognostic relevance of a positive or negative result. NCCN guidelines recommend SLN biopsy for all cutaneous melanoma patients with a primary tumor thickness greater than 1 mm and in selected patients with a thickness between 0.8 and 1 mm. However, they acknowledge a lack of consistency in its utility for prognosis and therapeutic value in tumors 1 mm and leave the decision to be made at the discretion of the patient and the attending physician.

For eligible individuals with CM, SLNB is the most reliable and accurate method of staging. In patients with CM, SLN status—whether positive or negative—is regarded as the crucial prognostic indicator for recurrence and the most reliable predictor of survival. Regarding adjuvant therapy and clinical trials, SLN status continues to be a reliable, independent predictor of outcome. The Working Group advises thorough discussion of the procedure risks and advantages with oncological surgery whenever possible for all patients eligible for SLNB. As previously indicated, SLNB provides exact staging, which can then encourage appropriate oncologist consultation and additional thought for adjuvant systemic medication or clinical trials.

Since SLNB was developed, many melanoma patients are no longer subjected to unnecessary LN dissections because they do not have a nodal disease. Staging is more precise, better predictive data is obtained, local disease control is improved, and there is strong evidence that patient survival is enhanced.

## 7. Future Directions

In clinical practice, adding predictive models, such as nomograms, is a recently suggested technique to help doctors assess each patient’s risk of SLN positivity in CM. However, as they need even more external validation in larger series, no such instrument or technique for evaluating the probability of nodal metastasis is yet to be included in management guidelines [76,77,78,79,80].

Recently, Lo et al. reported the creation of a nomogram based on tumor thickness, histological subtype, mitotic rate, and lymphovascular invasion to assess the rate of SLN metastasis. Faries et al. claim that despite variations in clinical and pathological parameters, this nomogram demonstrated higher rates of sensitivity and specificity compared to earlier models when it comes to predicting risk in both Australian and American cohorts. This shows not only that the tool might be used in populations all over the world but also that its increased accuracy could aid in the selection of patients for whom SLNB is unnecessary and would not be beneficial because they have a low likelihood (5%) of metastases [78]. Overall, these findings gave medical professionals insight into how to forecast SLN positives better, demonstrating whether and when it could be prudent to omit SLNB patients depending on clinicopathological traits [20].

Gene expression profiling is an alternative method of staging melanoma, and current research has examined and tested its utility in identifying SLN positivity [74]. The Decision Dx-Melanoma test, for instance, was designed to predict the risk of recurrence independent of conventional clinical and histological markers in patients with stage I-III melanoma. It is possible to establish whether the genetic profile of a specific tumor is closely connected to a low-risk or high-risk using a specialized gene expression profile test (31-GEP) and a predictive modeling method. Decision Dx-Melanoma may offer a different variable to assist doctors in identifying a viable patient population with a 5% probability of SLN positivity, as well as useful prognostic data on recurrence and disease survival, resulting in lower costs and resource optimization [20].

The inclusion of these models in clinical practice guidelines will require additional external validation across a larger sample size. Their predictive accuracy might be increased by including other pathological indicators, such as growth rate or speed rate. On the other hand, the future of melanoma staging will probably be liquid biopsies and gene expression profiling [81,82,83]. Furthermore, inhibiting melanogenesis in advanced CM could be a realistic strategy to enhance immune-, radio-, and chemotherapy [84]. In fact, a recent study proposed a prediction model that combined clinicopathological and gene expression factors and that, in some ways, can perform better than the Memorial Sloan Kettering Cancer Centre monogram [85].

## Figures and Tables

**Figure 1 medicina-58-01589-f001:**
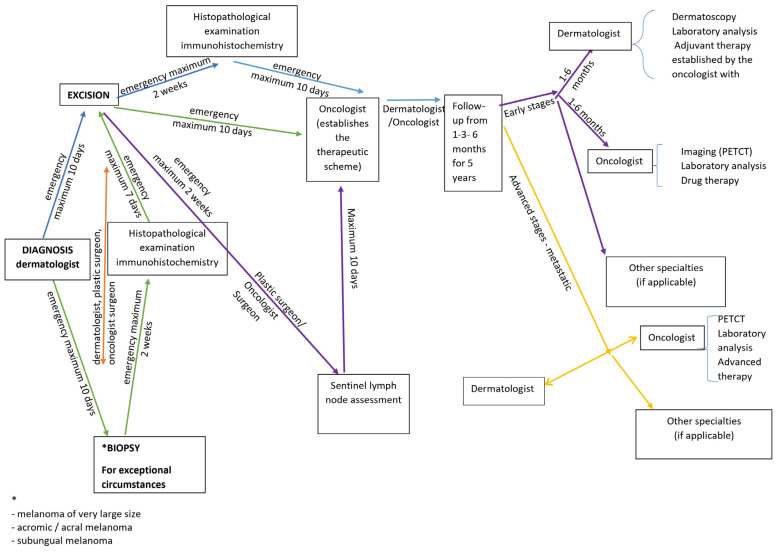
Romanian Society of Dermatology and Venereology algorithm for malignant melanoma.

## Data Availability

The data presented in this study are available on request from the corresponding author.

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
