# Peer review of "Sentinel Lymph Node Biopsy in Cutaneous Melanoma, a Clinical Point of View"

_medicina, 2022, doi:10.3390/medicina58111589_

Round 1
Reviewer 1 Report
Dear Authors
Your review is of outstanding quality. You did a very good job summarizing and presenting the state-of-the-art of the sentinel lymph node in cutaneous melanoma diagnosis, prognosis, and treatment.
However, I consider that an exhaustive review like this one should be accompanied by a well-designed figure that depicts the course of the disease and the surgical resection of a sentinel lymph node.
I recommend accepting after a minor revision, and this minor revision should be the addition of the figure mentioned.
Author Response
Dear Reviewer,
Thank you very much for your appreciations! Unfortunately, we consider that our clinical experience as dermatologists could not provide an exhaustive point of view for this surgical technique, that's why making a figure would be very difficult for us.
Kind regards,
Elena Porumb-Andrese
Reviewer 2 Report
The manuscript by Branisteanu et al. summarizes available information about sentinel lymph node biopsy in patients with cutaneous melanoma. The manuscript is overall well written, it is a nice collection old and new information; the authors point out limitations and contradictions, which altogether makes for a good quality of a manuscript. Minor comments are as listed:
· Basal cell carcinoma is the most frequent and squamous cell carcinoma is the second most frequent skin cancer. Melanoma is the most lethal, but hardly one of the most frequent skin malignancies. A source for different skin cancer prevalence should be provided after the first sentence.
· The references 1-4, used as sources for the first paragraph, are publications specifically about sentinel lymph node biopsy, while the paragraph does not mention the procedure. Proper manuscript with information described in the first paragraph should be provided.
· Similarly, for the guidelines listed in the second paragraph, source should also be provided.
· The authors mention the necessity to avoid “crucial errors” that could prohibit removal of proper SLNs. It would be good to elaborate what those crucial errors are. Also, if such data is available, a reason behind false-negative results should be explained.
· Clinical trials should be referenced with clinical trial number, or a publication if such is available, at the end of the sentence it was first cited. Listing all citations at the end of the paragraph is inconvenient for the reader who’s interested in following up on the data discussed.
· Section 3 is mislabeled as discussion. It should be given a proper title.
· Two comments about section 4: first, while the information provided by the authors is highly important, it is not the subject of this review as it has nothing to do with SLNs. Apart from that, title of the is worded incorrectly. In current form it means that melanoma patients have psychological and psychiatric impact on something. It should state (if it was to be used at all) “Psychological and psychiatric impact of melanoma”.
· While I leave this decision entirely to the authors discretion, I suggest changing the title of the manuscript. The present form - “are there any novelties?” has a negative connotation, somewhat discouraging the reader to take a closer look at the paper. If, however, the authors chose to stick to the title, I suggest clearly answering the question in the conclusion section.
Author Response
Dear Reviewer,
Thak you very much for offering us the opportunity of improving the quality of our manuscript! We made the canges according to the instructions.
Kind regards,
Dr. Porumb-Andrese